# High prevalence of malaria in pregnancy among women attending antenatal care at a large referral hospital in northwestern Uganda: A cross-sectional study

Caleb Mangusho[1], Edson Mwebesa[2], Jonathan Izudi[3], Mary Aleni[1], Ratib Dricile[4], Richard M. Ayiasi[4], Ismail D. Legason[4]*

1 Department of Nursing and Midwifery, Faculty of Health Sciences, Muni University, Arua, Uganda, 2 Department of Mathematics, Faculty of Science, Muni University, Arua, Uganda, 3 Department of Community Health, Faculty of Medicine, Mbarara University of Science and Technology, Mbarara, Uganda, 4 Department of Public Health, Faculty of Health Sciences, Muni University, Arua, Uganda

* id.legason@muni.ac.ug

## Abstract

### Background

Malaria in pregnancy contributes to substantial morbidity and mortality among women in Uganda. However, there is limited information on the prevalence and factors associated with malaria in pregnancy among women in Arua district, northwestern Uganda. We, therefore, assessed the prevalence and factors associated with malaria in pregnancy among women attending routine antenatal care (ANC) clinics at Arua regional referral hospital in north-western Uganda.

### Methods

We conducted an analytic cross-sectional study between October and December 2021. We used a paper-based structured questionnaire to collect data on maternal socio-demographic and obstetric factors and malaria preventive measures. Malaria in pregnancy was defined as a positive rapid malarial antigen test during ANC visits. We performed a modified Poisson regression analysis with robust standard errors to determine factors independently associated with malaria in pregnancy, reported as adjusted prevalence ratios (aPR) and 95% confidence intervals (CI).

### Results

We studied 238 pregnant women with a mean age of 25.32±5.79 years that attended the ANC clinic, all without symptomatic malaria. Of the participants, 173 (72.7%) were in their second or third trimester, 117 (49.2%) were first or second-time pregnant women, and 212 (89.1%) reported sleeping under insecticide-treated bednets (ITNs) every day. The prevalence of malaria in pregnancy was 26.1% (62/238) by rapid diagnostic testing (RDT), with the independently associated factors being daily use of insecticide-treated bednets (aPR

**Data Availability Statement:** All relevant data are within the paper and its Supporting information files.

**Funding:** The authors received no specific funding for this work.

**Competing interests:** The authors have declared that no competing interests exist.

0.41, 95% CI 0.28, 0.62), first ANC visit after 12 weeks of gestation (aPR1.78, 95% CI 1.05, 3.03), and being in the second or third trimester (aPR 0.45, 95% CI 0.26, 0.76).

## Conclusion

The prevalence of malaria in pregnancy among women attending ANC in this setting is high. We recommend the provision of insecticide-treated bednets to all pregnant women and early ANC attendance to enable access to malaria preventive therapy and related interventions.

## Introduction

Malaria is a significant public health problem in sub-Saharan Africa. Of the estimated 228 million global malaria cases in 2018, 93% were from sub-Saharan Africa [1]. Notwithstanding, 94% of the overall 405,000 malaria deaths in the same year were from sub-Saharan Africa [1]. Uganda has one of the highest malaria transmission rates in the region and is the 3rd most significant contributor to malaria global burden according to 2018 estimates [2]. Annually, malaria accounts for 30–50% of outpatient visits and one in every five reported deaths in Uganda [3].

Malaria in pregnancy is of particular concern and is associated with adverse birth outcomes [4, 5]. The less severe but chronic effects of malaria include maternal and newborn anemia, low birth weight, stunting, and reduced cognitive ability in children born to such mothers [6, 7]. The pathophysiological processes preceding adverse birth outcomes in pregnancy are initiated by parasite accumulation in placental intervillous spaces, causing inflammatory reactions and obstruction of nutrient supply to the fetus [8]. *Plasmodium falciparum (P. falciparum)* infection, in particular, is associated with very high placental parasite densities and the deposition of fibrinoid material [8]. It is this high placental parasitemia and accumulation of immune products that cause placental tissue damage and have the potential to cause adverse birth outcomes, including intrauterine fetal growth restriction, anemia, spontaneous abortions, preterm labor, low birth weight, and death [4, 5, 9, 10].

It is estimated that 10,000 maternal and 100,000 newborn deaths occur each year worldwide due to malaria, and nearly half of these cases are from Sub-Saharan Africa [11]. Malaria contributes to significant morbidity and mortality in Uganda. In 2019, 152 of the total 3,528 maternal and newborn deaths in Uganda were due to malaria in pregnancy [12]. The prevalence of malaria in pregnancy from population-based studies in Uganda ranges from 8.9% in the country's low transmission areas to over 50% in high transmission settings [13–15]. The risk factors for malaria in pregnancy are diverse and require context-specific interventions. Current evidence is that the risk of malaria is higher in the first pregnancy in younger pregnant women, and infections tend to peak early and decline towards term, reflecting the gradual acquisition of malaria immunity [16–18]. In a Malawian study, adolescent pregnant women (age<20) were more likely to suffer from malaria than older women, and the risk was highest in the early trimester [19].

Other risk factors associated with malaria in pregnancy include infection with human immunodeficiency virus (HIV) [20, 21], poor housing [15], low level of maternal education [15, 18, 21], and non-adherence to malaria preventive measures like insecticide-treated bed nets and IPTp [22, 23].

Uganda has made progress in implementing critical malaria control interventions, which include the provision of insecticide-treated bed nets, intermittent preventive treatment for pregnant women, prompt diagnosis and case management (test and treat), and indoor residual spraying in areas with high transmission intensity [24]. The national coverage of insecticide-treated bednets (ITNs) is reported at 83%, while IPTp has stagnated at 41% [25]. However, despite these efforts, there is no compelling evidence that the burden of malaria has decreased in recent years. Instead, some parts of the country have recently reported pockets of malaria 'epidemics' [26, 27]. The major factors that hinder the effective implementation of malaria in pregnancy control measures include low utilization of antenatal care (ANC) services leading to sub-optimal uptake of IPTp, improper case management leading to malaria rebound or even drug resistance, high transmission intensity, inadequate epidemic preparedness, and a limited epidemiological and surveillance data to inform decision making. In 2020, Uganda updated its guidelines for managing malaria in pregnancy following upsurges in 2018 and 2019 [28]. The new guidelines necessitate that all pregnant women attending routine ANC receive a test for malaria. Arua Regional Referral hospital was among the first regional hospitals to begin the implementation of the new guidelines for malaria in pregnancy. We used this opportunity to investigate the prevalence of malaria in pregnancy and its associated factors in this setting.

## Methods and materials

### Study design and setting

This was an analytic cross-sectional study conducted at the ANC clinic of Arua Regional Referral Hospital between October and December 2021. Arua hospital is located in Arua city and is the regional referral hospital for 12 other districts in north-western Uganda. The hospital serves about 3.5 million people in the region, including some parts of the neighboring Democratic Republic of Congo and South Sudan. The ANC clinic operates five days a week, Monday to Friday, from 8:00 am to 5:00 pm. On average, the clinic provides ANC services to approximately 120 women daily.

Mothers receive a focused antenatal care package which includes counseling, provision of ITNs, IPTp with sulfadoxine-pyrimethamine nutritional and personal hygiene education, obstetric scan, and sexually transmitted infections (STIs), malaria, anemia screening, ferrous and folic acid supplementation, and immunization. According to the Annual Health Sector Performance Report 2020/2021, only 31% of women attended the recommended four ANC visits, and IPTp coverage was below 43.2% in Arua [29].

Arua sits at an average elevation of 1310 meters above sea level, located between the drier East African plateau and the wetter Congo basin [30]. The climate is generally warm, with an average annual temperature of 23.0 °C [31]. The warmest month of the year is February, with an average temperature of 26.0°C, and the lowest average temperature occurs in August around 21.2°C [31]. The rainfall pattern is bimodal, with a dry season starting from mid-December to mid-March and a less predictable rainfall season occurring around March to June of every year [31]. The more reliable rainfall season begins in late July and until November each year [31]. The warm and relatively humid conditions favor the life cycle of *Anopheles gambiae*, the primary malaria vector in the region [32]. Arua district experiences perennial intense malaria transmission with asymptomatic parasite prevalence rates of 40–60% in school children [33, 34]. The annual entomological inoculation rate in the region is estimated at 397 infective mosquito bites per person per year, with *Anopheles gambiae* responsible for >80% of infective bites [35]. The peak transmissions occur around August to September and the lowest transmission intensities are observed yearly between January and March [35].

## Study population and sampling

All the pregnant women attending routine ANC clinics at Arua Regional Referral Hospital during the study period were eligible to participate in the study. We enrolled women attending any of the four ANC visits. At the same visit, blood samples were collected and analyzed. Pregnant women were excluded if they were critically ill or required hospitalization. We used a combination of systematic and simple random sampling to select the participants. Upon arrival, pregnant women were registered by the midwife, and basic biodata were collected. This information was used to construct a sampling interval. Using previous ANC pattern, we estimated that we would need ten days to enroll 238 participants at an average of twenty-four per day. We divided the number of women recorded in the ANC register by the number of women required for the interview on each day to obtain a sampling interval of five and henceforth, selected every 5th person recorded in the ANC register. Using the registration serial numbers, we then employed a simple random sampling method using a lottery approach to select the first and subsequent participants. This process was repeated daily until the required sample size was reached.

## Data collection and measurements

We enrolled participants in the second year (2021) of the Coronavirus disease (COVID-19) pandemic when all COVID-19 pandemic restrictions were lifted. Therefore, prenatal care and health service delivery was not affected. All the participants received a rapid malaria diagnostic test as a standard-of-care blood screening test. In this setting, rapid diagnostic tests (RDTs) are used as a point of care test partly because of the high clinic volumes but also for screening purposes. Therefore, no cases were tested by microscopy or other diagnostic methods. We used a paper-based structured questionnaire administered face-to-face by two trained research assistants. The questionnaire was a modified version of the malaria indicator survey tool, developed by the Monitoring and Evaluation Working Group of Roll Back Malaria. This international partnership coordinates global efforts to fight malaria [36]. The rationale for adopting this tool includes its replicability, and excellent psychometric properties, and exceptionally high reliability.

The outcome variable was malaria in pregnancy, measured by a positive malaria test result following a rapid malarial antigen test (RDT) as a standard of care at the hospital. The tests were performed by a laboratory technician with $\geq$ 10 years of work experience. Briefly, about 5μL of capillary or venous EDTA blood was introduced into the sample well using the sample transfer device that came with the RDT kit. This was followed by two drops of the assay buffer to the assay buffer well. The timer was set for 20 minutes before a test result could be read. The test results were interpreted anytime from 20 but before 30 minutes to minimize inaccurate test results that occur due to reaction product deterioration. A positive result was considered if two lines appeared on the cassette, otherwise it was negative if one control line appeared. In case a test line appeared without a control line or nothing at all appeared, the test was regarded invalid and a repeat test performed on a new device. Standard laboratory practices including aseptic procedures for obtaining samples and infection control measures were observed at all times by the laboratory personnel testing the samples. The performance of the test kits, SD Bioline Malaria Ag Pf (SD Bioline Inc. Korea) were as follows; sensitivity 98.2%, and specificity 91.6%, with positive and negative predictive values of 81.8% and 99.2% respectively [37].

The independent variables included age measured in absolute years later categorized as <25 versus $\geq$25 years, marital status namely married versus single, level of education categorized as none/primary versus secondary and beyond, and occupation measured as none/peasant versus employed/business. The age cutoff of 25 years was taken as a borderline to compare

malaria in pregnancy between early/young women and mature/older women. The other variables included gestational age categorized as the first trimester ($\leq$12 weeks) versus second/third trimester ($>$12 weeks), gravidity measured as $\leq$2 versus $\geq$3, time of first ANC visit measured as first trimester ($\leq$12) versus second/third trimester ($>$12 weeks), and time of intermittent preventive treatment of malaria in pregnancy start measured as early second trimester ($\leq$14 weeks) versus late trimester ($>$14 weeks).

We also collated data regarding knowledge of mothers about the intermittent preventive treatment of malaria in pregnancy doses (no versus yes), sleeping under insecticide-treated nets on daily basis (no versus yes), history of seeking obstetric care from a traditional birth attendant during the current pregnancy (no versus yes) and distance from home to a health facility measured as $\leq$5 km versus $>$5kms. Data on gestational age, use of insecticide-treated nets, use of intermittent preventive treatment of malaria in pregnancy, and recent ANC visits were verified using the ANC card at the time of the interview.

**Sample size estimation.** The study sample size was estimated using a formula by Kish Leslie [38]. We used a 16.9% prevalence of malaria in pregnancy among women from a previous study [39], a 95% confidence interval, and a 5% margin of error to obtain a sample size of 216 women. Sample size (n) = $Z^2p(1-p)/d^2$, where no = unadjusted sample size, Z = Z-score at 95% confidence level, p = proportion of the outcome in the population, and d = acceptable margin of error. Therefore, $n = 1.962x\ 0.169\ (1 - 0.169)/0.05^2 = 216$. We added a 10% non-response rate (22 participants) to the estimate to obtain an overall sample size of 238 participants.

**Data analysis.** Data were coded, cleaned, and entered in EpiData version 3.1 and then exported for analysis in Stata version 14 (Stata Corp, USA) statistical software. We descriptively summarized categorical data using frequencies and percentages and numerical data using mean and standard deviation. The prevalence of malaria in pregnancy was computed by dividing the number of participants with a positive test for malaria by the sample size, expressed as a percentage. The Chi-square test was used at the bivariate analysis level, to assess statistically significant differences between categorical variables and the outcome variable. Variables with p$\leq$0.2 at the bivariate analysis and those deemed clinically relevant from the literature, namely age of the mother, gravidity, use of insecticide-treated bed nets (ITNs), and intermittent preventive treatment of malaria in pregnancy (IPTp) were introduced in the model. Multivariable analysis was performed using a modified Poisson regression model with robust standard errors to avoid the violation of assumptions [40], reported as prevalence ratios (PR) and 95% confidence interval (CI). This analytic approach was appropriate because the outcome (malaria in pregnancy) was frequent, so using the odds ratio would overestimate the degree of association [41, 42]. We employed a stepwise backward regression analysis approach, excluding variables that did not improve the model fit based on the log-likelihood until model parsimony was reached. We checked for multicollinearity and assessed interaction and confounding at a 10% cut-off during model fitting.

## Quality control measures

To ensure data quality, the questionnaire was pretested on 10 participants outside the study area in August 2021 at another health facility located about two kilometers away from Arua Regional Referral Hospital. We trained the research assistants for two days regarding the study processes, responsible conduct of research, and questioning techniques. The regional laboratory participates in the ongoing national malaria external quality assessment (EQA) scheme and every quarter, a batch testing of RDTs is conducted prior to their issue for laboratory use. The regional external quality assurance team samples RDT cassettes from each batch and are

tested using a pair of known positive and negative blood samples confirmed by expert micros-copy and PCR. Therefore, the batch used for malaria testing in the present study passed QC testing. The lab technician performed initial testing of the patients and results were reviewed by the research assistant and entered into appropriate data collection forms. The research assistants were supervised by a research team lead who reviewed the filled questionnaires daily for completeness and accuracy. Concerning data on obstetric history, the completed question-naire was verified against the ANC register or card.

### Ethical considerations

This study received approval from the Lacor Hospital Institutional Research Ethics Committee (LHIREC#2021–61), Uganda National Council of Science and Technology (HS2210ES). It was further granted administrative approval by the Arua Hospital administration. We sought voluntary informed written consent from pregnant women aged ≥18 years. For participants below the age of 18 years, an adult either a spouse or guardian provided written informed con-sent in addition to assent from the participant. The research assistant explained the study's purpose, risks, and benefits in the local language and provided ample time for participants to ask questions/clarifications. The participants were further assured of anonymity and confi-dentiality of the data collected. Pregnant women were free to refuse to participate or withdraw from the study without any prejudice to services offered at the clinic.

## Results

### Background characteristics of pregnant women and prevalence of malaria in pregnancy

We studied 238 women with a mean age of 25.32±5.79 years who had attended the ANC clinic, and all were not malaria symptomatic cases. Of these, 207 (87.0%) resided within a 5 km radius of the health facility, 191 (80.3%) were married, 149 (62.6%) attained at least a sec-ondary level of education, and 183 (76.9%) were unemployed. Also, 173 (72.7%) were in their second or third trimester, and 117 (49.2%) were pregnant for the first or second time. The majority of the women, 169 (71.0%), started ANC in the first trimester, 71 (30.1%) began IPTp late (> 14 weeks of pregnancy), 137 (57.6%) had adequate knowledge about the correct IPTp doses, 212 (89.1%) reported sleeping under ITNs every day, and 111 (46.6%) reported seeking obstetric care from a traditional birth attendant (TBA) during the current pregnancy. Overall, 62 (26.1%) of the participants had malaria in pregnancy by rapid diag-nostic tests (RDT), with all the malaria-positive cases (62 or 100.0%) due to *P. falciparum* infection.

In Table 1, our data show that malaria in pregnancy was prevalent in the first trimester compared to the second or third trimester (35.4% versus 22.5%), and among women who attended less than four ANC visits compared to those who had attended ≥4 ANC visits (74.4% versus 71.0%, respectively). However, malaria in pregnancy was less common among pregnant women aged <25 years compared to those aged ≥25 years (25.8% versus 26.3%, respectively), those in the category of one or two pregnancies versus three or more (24.8% versus 27.3%, respectively), those that had ANC visit in the first trimester (≤12 weeks) versus later trimester (>12 weeks) (23.7% versus 31.9%, respectively) and those that had early IPTp initiation (≤14 weeks) versus late IPTp initiation (>14 weeks) (23.6% versus 32.4%, respectively). Being in the first trimester (*p = 0.044*), daily ITN use (*p<0.001*), and seeking obstetric care from a tradi-tional birth attendant (*p = 0.036*) showed significant differences in malaria in pregnancy in the bivariate analysis.

**Table 1. Background characteristics of pregnant women and prevalence of malaria in pregnancy.**

| Variables | Overall, N (%) | Malaria in pregnancy | | p-value |
| --- | --- | --- | --- | --- |
| | | Positive N (%) | Negative N (%) | |
| Age (years) | | | | |
| Below 25 years | 120 (50.4) | 31 (25.8) | 89(74.2) | |
| 25 years and above | 118 (49.6) | 31 (26.3) | 87 (73.7) | 0.939 |
| Mean (SD) | 25.32 (5.79) | | | |
| Marital status | | | | |
| Married | 191 (80.3) | 48 (25.13) | 143 (74.9) | |
| Single | 47 (19.7) | 14 (29.8) | 33 (70.2) | 0.515 |
| Level of education | | | | |
| Primary or none | 89 (37.4) | 28 (31.5) | 61 (68.5) | |
| Secondary and beyond | 149 (62.6) | 34 (22.8) | 115 (77.2) | 0.142 |
| Type of job | | | | |
| Housewife or peasant | 183 (76.9) | 51 (27.9) | 132 (72.1) | |
| Employed or business | 55 (23.1) | 11 (20.0) | 44 (80.0) | 0.244 |
| Distance from home to health facility (km) | | | | |
| $\leq 5$ | 207 (87.0) | 56 (27.1) | 151 (72.9) | |
| >5 | 31 (13.0) | 6 (19.4) | 25 (80.6) | 0.362 |
| Gestational age (in weeks) of current pregnancy | | | | |
| $\leq$12 weeks | 65 (27.3) | 23 (35.4) | 42(64.6) | |
| >12 weeks | 173 (72.7) | 39(22.5) | 134(77.5) | 0.044 |
| Number of pregnancies (gravidity) | | | | |
| One or two pregnancies | 117 (49.2) | 29 (24.8) | 88 (75.2) | |
| Three or more | 121 (50.8) | 33 (27.3) | 88 (72.7) | 0.662 |
| Time of first ANC visit | | | | |
| 1st trimester | 169 (71.0) | 40 (23.7) | 129 (76.3) | |
| 2nd or 3rd trimester | 69 (29.0) | 22(31.9) | 47 (68.1) | 0.190 |
| Number of ANC Visits | | | | |
| < 4 Visits | 207 (87.0) | 154 (74.4) | 53 (25.6) | |
| $\geq$ 4 Visits | 31 (13.0) | 22 (71.0) | 9 (29.0) | 0.685 |
| Time of IPTpstarted | | | | |
| Early 2nd trimester ($\leq$14 Weeks) | 165 (69.9) | 39 (23.6) | 126 (76.4) | |
| Late (> 14 Weeks) | 71 (30.1) | 23 (32.4) | 48 (67.6) | 0.161 |
| Knowledgeable about IPTp doses | | | | |
| No | 101 (42.4) | 31 (30.7) | 70(69.3) | |
| Yes | 137 (57.6) | 31 (22.6) | 106(77.4) | 0.161 |
| Sleeping under ITN daily | | | | |
| No | 26 (10.9) | 15 (57.7) | 11 (42.3) | |
| Yes | 212 (89.1) | 47 (22.2) | 165 (77.8) | <0.001 |
| Seeking obstetric care from a TBA | | | | |
| No | 127 (53.4) | 26(20.5) | 101(79.5) | |
| Yes | 111 (46.6) | 36 (32.4) | 75 (67.6) | 0.036 |

## Factors associated with malaria in pregnancy

The unadjusted and adjusted analysis results are presented in Table 2. In the unadjusted analysis, malaria in pregnancy was less likely among women who everyday slept under insecticide-treated bed nets (unadjusted prevalence ratio (PR), 0.38; 95% CI, 0.25–0.58) and those in the

**Table 2. Modified Poisson regression analysis of factors associated with malaria in pregnancy.**

| Covariates | Categories | Overall, N (%) | Unadjusted PR (95% CI) | Adjusted PR (95% CI) |
|---|---|---|---|---|
| Age (years) | Below 25 years | 120 (50.4) | 1 | 1 |
| | 25 years and above | 118 (49.6) | 1.02 (0.66, 1.56) | 1.05 (0.67, 1.62) |
| Sleeping under ITN daily | No | 26 (10.9) | 1 | 1 |
| | Yes | 212 (89.1) | 0.38 (0.25, 0.58) * | 0.41 (0.28, 0.62) *** |
| Seeking obstetric care from a TBA | No | 127 (53.4) | 1 | 1 |
| | Yes | 111 (46.6) | 1.58 (1.02, 2.45) * | 1.55 (0.99, 2.43) |
| Mother's educational level | Primary or none | 89 (37.4) | 1 | |
| | Secondary & above | 149 (62.6) | 0.73(0.47, 1.11) | |
| Knowledge about IPTp doses | No | 101 (42.4) | 1 | |
| | Yes | 137 (57.6) | 0.74 (0.48, 1.13) | |
| Time of first ANC visit | First trimester | 169 (71.0) | 1 | 1 |
| | Second or third trimester | 69 (29.0) | 1.35 (0.87, 2.09) | 1.78 (1.05, 3.03) * |
| The time of IPTp started | Early second trimester (13 or 14 Weeks) | 165 (69.9) | 1 | |
| | Late (> 14 Weeks) | 71 (30.1) | 1.37 (0.89, 2.12) | |
| Number of pregnancies (gravidity) | One or two pregnancies | 117 (49.2) | 1 | |
| | Three or more | 121 (50.8) | 1.10 (0.72, 1.69) | |
| Gestational age in weeks (current pregnancy) | ≤12 weeks | 65 (27.3) | 1 | 1 |
| | >12 weeks | 173 (72.7) | 0.64 (0.41, 0.98) | 0.45 (0.26, 0.76) ** |
| Model summary | Akaike Information Criteria (AIC) | | | 284.4 |
| | Deviance, the goodness of fit | | | 148.4 |
| | | | | (p = 1.000) |

Note: 1) Statistical significance at 5%:

*p<0.05

**p<0.01

***p<0.001;

PR: Prevalence ratio

second or third trimester (PR 0.64; 95% CI, 0.41–0.98). But it was more likely among women who had initiated ANC late (>12 weeks) (PR1.35; 95% CI, 0.87–2.09), and among women who sought obstetric care from a traditional birth attendant during the current pregnancy (PR, 1.58; 95% CI, 1.02–2.45).

In the multivariable or adjusted analysis, the factors that remained statistically significant were: sleeping daily under ITNs (adjusted prevalence ratio (aPR), 0.41; 95% CI 0.28, 0.62), gestational age (aPR 0.45, 95% CI 0.26, 0.76), and time of first ANC visit (aPR, 1.78; 95% CI 1.05, 3.03). The relationship between the variables seeking obstetric care from a TBA and malaria in pregnancy was confounded by maternal age in the multivariable analysis. The other variables namely: knowledge about IPTp doses, time of IPTp started, number of ANC visits, and gravidity were not statistically significant in the final model (Table 2).

## Discussion

This study investigated the prevalence and factors associated with malaria in pregnancy in a region where malaria transmission is holo-endemic. The overall prevalence of malaria in

pregnancy in our study was 26.1% by rapid diagnostic test (RDT). Since none of the participants was symptomatic of malaria, our results represent the burden of asymptomatic malaria in pregnancy in this region. The significance of asymptomatic *P. falciparum* infections, particularly during pregnancy, cannot be overemphasized. Asymptomatic *P. falciparum* infection, if not promptly treated, may progress to chronic parasitemia, characterized by placental sequestration of infected red blood cells [8, 43]. The accumulation of infected red blood cells and parasite-activated immune products have the potential to cause placental tissue damage and dyserythropoiesis (defective development of red blood cells), leading to maternal anemia and other associated adverse birth events, namely premature delivery, stillbirth, and intrauterine fetal growth restriction among others [4, 9, 10]. Our study has shown a high prevalence of malaria in pregnancy compared to a previous study by Braun *et al.* [13] who report an 8.9% malaria prevalence among pregnant women in Western Uganda, and a 13.9% prevalence of malaria in pregnancy reported by Namusoke et al. [14] in central Uganda but a lower than 51.1% reported by Okiring *et al.* [15] in Eastern Uganda. Generally, our results compare favorably to similar findings across sub-Saharan Africa [17, 44, 45].

The differences in the prevalence of malaria in pregnancy in our study and other similar studies can be attributed to differences in study settings, socio-economic and diagnostic approaches. Peak malaria transmission rates occur between August and September in the West Nile region [35]. Unlike the previous studies that recruited for a more extended period (at least six months), our study recruited over three months from October to December, a period characterized by moderate malaria transmission in the study setting [35]. Therefore, our study may have underrepresented the burden of malaria in pregnancy in this region. Another essential factor to consider is geographic variations. Malaria is holoendemic in more than 95% of the country [24], albeit some parts are experiencing unstable malaria transmission [3]. For instance, parts of central Uganda, Mid-Eastern, and South West experience some of the lowest prevalences of malaria, while East Central, North East, and West Nile have one of the country's highest prevalences approaching 50.0% [3, 35, 46].

The relationship between socioeconomic gradient and infectious disease risk has been well documented [47]. Poor individuals or communities are less likely to have suitable housing and equitable access to health care. Our data show that up to 76.9% of the pregnant women neither owned any form of business nor were employed, and 37.4% had not received any formal education. The national household survey results of 2019/2020 indicated that the poverty levels in the West Nile region reduced dramatically from 35% in 2017 to 17% in 2019 [48]. However, compared to Central and Western regions where the previous studies were conducted, the West Nile region ranks below key dimensions of the Human Development Index (HDI), a measure of a country's development concerning healthcare, education, and life expectancy [49]. For example, poverty levels in the Kampala region were estimated at 2%, and it was 17% in West Nile [48]. The corresponding adult literacy rates among women in these two regions were estimated as 89.1%, and 59.2%, respectively [48]. These variations influence the prevalence of malaria in pregnancy.

To establish malaria infection, we used rapid diagnostic tests. While the previous studies used blood smear microscopy, histology, or polymerase chain reaction (PCR)-based molecular methods [14, 15]. Malaria rapid diagnostic tests can be instrumental in making a quick diagnosis of malaria in circumstances where microscopy-based diagnoses may be unavailable or unreliable. In the study setting, rapid diagnostic tests are widely used as a point of care test in high-volume units such as antenatal clinics and outpatient departments to counter the workload and partly for screening purposes. The limitations of rapid diagnostic tests have been well described [50] and they include low sensitivity compared to microscopy when parasitemia is <100 parasites/μL, a false negative result due to parasite genetic variability, and prozone effect

[51]. Additionally, more than 80% of the rapid diagnostic tests (RDTs) used in Uganda are histidine-rich protein (HRP2)-based, an antigen-specific for *Plasmodium falciparum species* [50]. Although this species accounts for ≥95% of malaria infections in Uganda [50], there is a risk of missing non-*Plasmodium falciparum* infections. Theoretically, rapid diagnostic tests are inferior compared to blood smear microscopy, placental histology, and polymerase chain reaction (PCR)-based methods.

Our data show that malaria in pregnancy is less likely among women who everyday slept under an insecticide-treated bed net compared to those who never slept under an insecticide-treated bed net every day. This finding is consistent with that of Fana et al. [17] in Nigeria and Cisse et al. [18] in Burkina Faso. Reports of consistent Insecticide-treated bednet (ITN) use have shown that ITNs are highly effective in reducing malaria incidence. For example, a Malawian study found a 30% reduction in malaria incidence among children less than 5 years after initial household ITN distribution in the study area [52]. In an earlier systematic review, ITNs compared to no nets showed a positive impact on birthweights, and reduced stillbirths/miscarriages by 33%, while placental parasitemia was reduced by 23% in all gravidae [53]. It has also been observed that a high coverage and use of insecticide-treated bednets (ITNs) can potentially reduce mosquito vector density resulting in broader community protection against malaria [25]. However, challenges of misuse of ITNs and lack of proper education regarding care and maintenance for ITNs may reduce their protective effects. For example exposure of ITNs to ultraviolet light (UV), washing them with detergents, and frequency of washing are all important determinants of ITN longevity [54]. Therefore, mass ITN distribution campaigns should be strengthened with the appropriate social behavioral change and communication (SBCC) programs to promote good net use at the household level.

Our finding that malaria in pregnancy is more likely among women who initiated antenatal care (ANC) late (attended the first ANC in the second or third trimester) compared to those who attended the first ANC visit in the first trimester is consistent with previous findings of Anchang-Kimbi et al. [55] and Nkoka et al. [56]. Early ANC attendance promotes the uptake of malaria prevention services including the use of insecticide-treated bednets (ITNs) and intermittent preventive treatment of malaria in pregnancy (IPTp). During antenatal care, women also receive health education on household sanitation, care and proper use of insecticide-treated bednets and early health seeking behaviour. All this may contribute to reduction in the risk of malaria among women who initiate early antenatal care.

Relatedly, the finding that malaria in pregnancy is less likely in the second or third trimester is consistent with the findings of Simon-Oke et al.[45] in Nigeria, Accrombessi et al. [57] in Benin, and Gontie et al. [58] in Ethiopia. The biologically plausible reason is the lack of protection against malaria during the first few months of pregnancy due to systematic cytokine bias, which leads to weakened immunity [59, 60]. Yet, it is also plausible that women may contract malaria before becoming pregnant, which would explain the increased prevalence seen in the first trimester. It would be interesting to compare malaria positivity at the first prenatal appointment with the positivity of the test during subsequent visits, even if our study did not analyse data from two time points to establish this connection.

Although we did not find a significant relationship between maternal age and malaria in pregnancy prevalence in our study, several studies have reported an association between young maternal age (<20 years) and malaria in pregnancy [19, 61–63]. The lack of association in our study could be explained, in part, by the fact that a small proportion (18%) of the women studied were below the age of 20 years and this may have reduced the power to detect any significant differences. The association between young maternal age and malaria in pregnancy is thought to be related to the level of acquired malaria immunity which according to literature increases with the frequency of infections during successive pregnancies [61–63]. It

could also mean that younger women are more likely than older ones to become first-time mothers, and as a result, they may be at a higher risk of contracting malaria while pregnant than older ones who may have had many infections during their pregnancies.

## Study strengths and limitations

Our study has several strengths and limitations. First, we selected a random sample from a large tertiary care facility which is more likely to be heterogeneous. Hence, our results can be generalized to the population under study. Second, we controlled for interviewee response bias by verifying self-reported data against ANC records. Third, we used the malaria indicator survey toolkit, a validated tool with excellent psychometric properties, to collect all the data in the study. Our study had limitations that should be considered in the interpretation of the findings, and they include 1) A relatively small sample, which might have reduced the statistical power to detect a true association in the population; 2) The study recruitment period was relatively short (October to December or three months), and it coincided with moderate malaria transmission intensity [35]. The malaria transmission intensity in our setting peaks around August and September every year [35], so the study might have underestimated the burden of malaria in pregnancy; 3) The ANC coverage in our setting was approximately 30.0%, which is lower than the recommended ≥4 ANC visits [29], implying that a high percentage of potentially malaria-infected pregnant women had missed the opportunity to participate in the study. This might have led to an inaccurate estimate of the prevalence of malaria in pregnancy in the study; 4) The study did not include symptomatic malaria cases, and more importantly so there is the possibility that the pregnant women who sought ANC are likely those most protected from malaria and other diseases; 5) The study lacked explanatory variables, such as the reasons for not sleeping under an insecticide-treated bednets daily among the participants as these were not within the scope of the study; and, 6) malaria in pregnancy was diagnosed with a low-sensitivity method so some cases might have been erroneously classified as uninfected if the parasite densities were below the detection limits. Rapid diagnostic assays for malaria based on the histidine-rich protein (HRP-2) have sensitivity and specificity that vary from 95% to 98.2% and 59.2 to 86.3%, respectively, in investigations including general populations [37, 64–66]. However, due mostly to placental parasite sequestration, the tests are typically less sensitive when used for malaria diagnosis in pregnant women [67, 68]. We admit that despite the fact that it was outside the scope of our study to evaluate the performance of histidine rich protein (HRP2)-based rapid diagnostic tests (RDTs), this may have contributed to the study's low rate of malaria in pregnant women. Not to mention, we recognize the limitations brought on by the choice of our cut-off points for variables like age and gravidity. They further restrict the comparability of our findings with previous studies investigating teenage pregnancy and primigravida as potential risk factors for malaria in pregnancy. Nevertheless, our findings are insightful and should be carefully interpreted to direct malaria interventions among pregnant women.

## Implications of the study findings

Our findings have ramifications for clinical care, public health prevention, and control, including surveillance and future research. Our study has shown that malaria in pregnancy is prevalent, necessitating routine surveillance for malaria at the individual and population levels. On the clinical front, our findings emphasize a need to routinely test pregnant women for malaria to enable early malaria diagnosis and treatment, thereby preventing adverse pregnancy and birth outcomes. Our study also provides a benchmark for future research about malaria in pregnancy in the region. Few studies have focused on malaria in pregnancy in the West Nile

region, so our findings might inform the design of future observational and interventional studies. Policy-wise, the enforcement of malaria prevention and control measures at the family and community levels is important to reduce the burden of malaria in the region. The use of cost-effective and proven malaria prevention measures such as insecticide-treated bed nets (ITNs) and intermitted preventive therapy in pregnancy (IPTp) should be promoted. Environmental control measures such as the draining of stagnant waters on the compound, cleaning/slushing busy areas, and discarding broken/empty containers that provide breeding grounds for mosquitoes have to be enforced.

## Conclusions and recommendations

Our study shows that malaria in pregnancy is high among women seeking ANC at Arua Regional Referral Hospital. The study further shows that consistent ITN use and early ANC initiation may significantly reduce the risk of malaria in pregnancy. We recommend the provision of insecticide-treated bed nets to all pregnant women and early ANC attendance to enable access to malaria preventive therapy and related interventions to prevent malaria in pregnancy.

## Supporting information

**S1 Table. Background characteristics of pregnant women and prevalence of malaria in pregnancy.**
(TIF)

**S2 Table. Modified Poisson regression analysis of factors associated with malaria in pregnancy.**
(TIF)

**S1 Checklist. STROBE checklist.**
(TIFF)

**S1 File. Analysis file.**
(CSV)

**S1 Dataset.**
(ZIP)

## Acknowledgments

We thank the pregnant women who participated in the study. We are grateful to the hospital staff, particularly the nurses and midwives working in the ANC clinics who helped with data collection and availed the ANC registers, and made it possible to interview their clients. We also want to express our gratitude to the hospital management for permission to conduct the study.

## Author Contributions

**Conceptualization:** Caleb Mangusho, Ismail D. Legason.

**Data curation:** Caleb Mangusho, Jonathan Izudi, Ismail D. Legason.

**Formal analysis:** Edson Mwebesa, Jonathan Izudi, Ismail D. Legason.

**Investigation:** Ismail D. Legason.

**Methodology:** Edson Mwebesa, Jonathan Izudi, Ismail D. Legason.

**Project administration:** Mary Aleni.

**Resources:** Mary Aleni, Ratib Dricile, Ismail D. Legason.

**Software:** Edson Mwebesa.

**Supervision:** Caleb Mangusho, Mary Aleni, Ratib Dricile, Ismail D. Legason.

**Writing – original draft:** Ismail D. Legason.

**Writing – review & editing:** Jonathan Izudi, Ratib Dricile, Richard M. Ayiasi.

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
