## [Decision Letter · Decision Letter 0]

30 Jun 2022

PONE-D-22-12136High prevalence of malaria in pregnancy among women attending antenatal care at a large referral hospital in northwestern Uganda: a cross-sectional studyPLOS ONE

Dear Dr. Legason,

Thank you for submitting your manuscript to PLOS ONE. After careful consideration, we feel that it has merit but does not fully meet PLOS ONE’s publication criteria as it currently stands. Therefore, we invite you to submit a revised version of the manuscript that addresses the points raised during the review process.Need to strengthen the Introduction with relevant references Suggest to improve the Discussion with implication of the study in UgandaPlease submit your revised manuscript by Aug 14 2022 11:59PM. If you will need more time than this to complete your revisions, please reply to this message or contact the journal office at plosone@plos.org. Please include the following items when submitting your revised manuscript:A rebuttal letter that responds to each point raised by the academic editor and reviewer(s). You should upload this letter as a separate file labeled 'Response to Reviewers'.A marked-up copy of your manuscript that highlights changes made to the original version. You should upload this as a separate file labeled 'Revised Manuscript with Track Changes'.An unmarked version of your revised paper without tracked changes. You should upload this as a separate file labeled 'Manuscript'.

We look forward to receiving your revised manuscript.

Kind regards,

Thae Maung Maung, MBBS, MSc (International Health), PhD

Academic Editor

PLOS ONE

Journal Requirements:

Reviewers' comments:

Reviewer's Responses to Questions

**Comments to the Author**

1. Is the manuscript technically sound, and do the data support the conclusions?

Reviewer #1: Partly

Reviewer #2: Yes

Reviewer #3: Partly

2. Has the statistical analysis been performed appropriately and rigorously? 

Reviewer #1: No

Reviewer #2: I Don't Know

Reviewer #3: Yes

3. Have the authors made all data underlying the findings in their manuscript fully available?

Reviewer #1: Yes

Reviewer #2: Yes

Reviewer #3: No

4. Is the manuscript presented in an intelligible fashion and written in standard English?

Reviewer #1: No

Reviewer #2: Yes

Reviewer #3: No

5. Review Comments to the Author

Reviewer #1: The text needs revision in its length and standardization in the use of parentheses, for example when interleukins are cited in line 67 of the text.

Additionally, on the results presented in absolute numbers and percentages, all calculations must be redone because some contain errors, for example, in lines 204 to 205.

In the data analysis, some cut-off points in the conversion of some numeric variables into dichotomies are not clear, such as, for example, the variable age in 25 years; it is known that susceptibility and vulnerability to malaria in pregnant women is greater in adolescents and primigravidae. Of those who were on their first prenatal visit what was the prevalence?

The study describes the prevalence and factors associated with having a positive rapid test for malaria during pregnancy in Uganda. The recruitment of research participants took place during the months of October to December 2021. There is no further information about:

1. The season in which the study is carried out and how it influences malaria transmission.

2. How does prenatal control work and what is the coverage in the study area?

3. Considering that the samples were collected during the second year of the Covid-19 pandemic, how the pandemic may have influenced the results in terms of altering prenatal coverage and the functioning of malaria control programs such as distribution of impregnated mosquito nets and intermittent preventive treatment.

Reviewer #2: This study revealed a high prevalence of malaria during pregnancy in women attending the antenatal care (ANC) clinics at the Arua Regional Referral Hospital. Beyond that, the paper showed the importance, in this endemic setting, of the daily use of insecticide-treated bed nets and early ANC attendance. Therefore, I believe this is a valuable paper for the scientific community that may contribute to base public health policies in Arua or comparable endemic regions.  

I have some minor observations that I hope could be helpful for the authors. 

1. Introduction lines 62-71: Rather than the biology of gestational malaria, it may be more interesting to use the introduction to delve deeper into the epidemiology of malaria in pregnancy and malaria control measures. For example: have malaria control measures in Uganda changed over time? What are the weaknesses that impair the wide implementation and adherence of the intermittent preventive treatment of malaria in pregnancy in Uganda? How does malaria diagnosis and treatment work in Uganda? 

2. Methods and materials lines 116 - 117: I suggest specifying whether you included both pregnant women who went to the clinic for routine ANC and pregnant women who went to the clinic because they felt sick. Does the clinic attend women attending for routine ANC and pregnant women with suspected malaria or other diseases? I think it would be interesting to specify the number of ANC visits during pregnancy recommended in Uganda (four?) and the adherence to this recommendation (if you have that information). 

3. Lines 132-133: It is important to specify the Plasmodium species. The diagnosis was done with blood smear microscopy too? 

4. Results line 209: If you have information on the proportion of symptomatic and asymptomatic malaria cases, it may be interesting to include it. 

5. Lines 146 and 270: Sleeping without insecticide-treated nets on daily basis means no nets, or no nets or untreated nets? If you have that information it may be interesting to describe it in the discussion. 

6. Line 231: I suggest adding one more column with the n of each category (for example "n with malaria/n total"), even though this information has already been placed in table 1. 

7. Line 306: Regarding the limitations of the study, is it possible that the population most exposed to malaria has less access to the health facility? In this case, this study would not be overestimating the malaria cases.  

Finally, consider adding a STROBE guideline chart as a supplementary file.

Reviewer #3: Summary of the research and overall impression

This manuscript aimed to identify the prevalence of MiP and factors associated with MiP among pregnant women in Western Uganda. Introduction should be strengthen with more literature review and presenting the burden of malaria and MiP in Uganda. Methodology and analysis is reasonable. Results session needs revision in text description and background variables analysis should be revisited. Discussion is very weak and required major revision. It needs to improve in terms of approach to discussion, referencing and citation, literature review and linked to the key findings, etc. Academic and scientific English language in presenting and discussing throughout the manuscript is also weak and required proper langue editing.

Please find the detail comments in the attached document.

6. PLOS authors have the option to publish the peer review history of their article (what does this mean?). If published, this will include your full peer review and any attached files.

Reviewer #1: No

Reviewer #2: No

Reviewer #3: **Yes: **Poe Poe Aung

---

## [Author Response · Author response to Decision Letter 0]

14 Aug 2022

Review Comments to the Author

Reviewer #1:

1. The text needs revision in its length and standardization in the use of parentheses, for example when interleukins are cited in line 67 of the text.

Author response: Thank you. We have revised the section and it reads: 

“The pathophysiological processes preceding adverse outcomes in pregnancy are initiated by parasite accumulation in placental intervillous spaces, causing inflammatory reactions and obstruction of nutrient supply to the fetus. Plasmodium falciparum infection, in particular, is associated with very high placental parasite densities and deposition of fibrinoid material. It is this high placental parasitemia and accumulation of immune products that cause placental tissue damage and have the potential to cause adverse birth outcomes including Intra uterine fetal growth restriction (IUGR), anemia, spontaneous abortions, preterm labor, low birth weight and death”

2. Additionally, on the results presented in absolute numbers and percentages, all calculations must be redone because some contain errors, for example, in lines 204 to 205.

Author response: We have re-analyzed the data and have reported the results as follows: 

“We studied 238 women with a mean age of 25.32±5.79 years and all were attending the clinic for routine ANC, no malaria symptomatic cases were recorded. Of these, 87% (n=238) resided within a 5 km radius of the health facility, 80.3% (n=238) were married, 62.6% (n=238) attained at least a secondary level of education, and 76.9%(n=238) were unemployed. One hundred seventy-three, representing 72.7%(n=238) of the pregnant women were in their second or third trimester and about half (49.2%, n=238) had one or two pregnancies. The majority of the women (71%, n=238) started ANC in the first trimester, 30.1% (n=236) started IPTp late (> 14 weeks of pregnancy) and 89.1% (n=238) reported sleeping under ITNs every day”.

3. In the data analysis, some cut-off points in the conversion of some numeric variables into dichotomies are not clear, such as, for example, the variable age in 25 years; it is known that susceptibility and vulnerability to malaria in pregnant women is greater in adolescents and primigravidae. Of those who were on their first prenatal visit what was the prevalence?

Author response: We have corrected this and provided an explanation detailed below:

“The age cutoff of 25 years was taken as a borderline to compare MiP between early/young women versus mature/older women” 

4. The study describes the prevalence and factors associated with having a positive rapid test for malaria during pregnancy in Uganda. The recruitment of research participants took place during the months of October to December 2021. There is no further information about:

1. The season in which the study is carried out and how it influences malaria transmission.

Author response: This has been addressed as follows:

 “Arua sits at an average elevation of 1310 meters above sea level, located between the drier East African plateau and the wetter Congo basin. The climate is generally warm with an average annual temperature of 23.0 ºC. The warmest month of the year is February with an average temperature of 26.0 ºC and the lowest average temperature in the year occurs in August when it is around 21.2ºC. The rainfall pattern is bimodal with a dry season starting from mid-December to mid-March and a less predictable rainfall season occurs around March to June of every year. The more reliable rainfall season begins in late July until November each year. The warm and relatively humid conditions favor the life cycle of Anopheles gambiae, the main malaria vector in the region . Arua district experiences perennial intense malaria transmission with asymptomatic parasite prevalence rates of 40- 60% in school children.The annual entomological inoculation rate in the region is estimated at 397 infective mosquito bites per person per year with Anopheles gambiae responsible for >80% of infective bites. The peak transmissions occur around August to September and the lowest transmission intensities are observed between January and March every year. 

”

2. How does prenatal control work and what is the coverage in the study area?

Author response: We have added a new sentence and it reads: 

“The ANC clinic operates 5 days a week, Monday to Friday, from 8:00 am to 5:00 pm. The clinic provides ANC services to approximately 120 women per day, on average. Mothers receive focused antenatal care package which includes counseling, provision of ITNs, IPTp with sulfadoxine-pyrimethamine, nutritional and personal hygiene education, obstetric scan, sexual transmitted infections (STIs), malaria, anemia screening, ferrous and folic acid supplementation, and immunization. According to the Annual Health Sector Performance Report 2020/2021, only 31% of women attended the recommended four ANC visits and IPTp coverage is below 43.2% in Arua”

5. Considering that the samples were collected during the second year of the Covid-19 pandemic, how the pandemic may have influenced the results in terms of altering prenatal coverage and the functioning of malaria control programs such as distribution of impregnated mosquito nets and intermittent preventive treatment.

Author response: By October last year, most sectors of the economy were operating. Public transport was opened and there were no movement restrictions. Additionally, the government conducted a mass ITN distributed between 2020/2021 despite the covid-19 restrictions. Community health workers reached every home and provided home-based delivery of ITNs. Therefore, prenatal care and health service delivery was not affected by the pandemic as restrictions had ceased. For example, the daily ANC attendance during our study was not significantly different compared to the pre-Covid-19 periods. 

We have therefore added a new sentence and it reads:

“We enrolled participants in the second year (2021) of the Coronavirus disease (COVID-19) pandemic, a time when all COVID-19 pandemic restrictions were lifted. Therefore, prenatal care and health service delivery was not affected.”

Reviewer #2: 

This study revealed a high prevalence of malaria during pregnancy in women attending the antenatal care (ANC) clinics at the Arua Regional Referral Hospital. Beyond that, the paper showed the importance, in this endemic setting, of the daily use of insecticide-treated bed nets and early ANC attendance. Therefore, I believe this is a valuable paper for the scientific community that may contribute to base public health policies in Arua or comparable endemic regions. 

I have some minor observations that I hope could be helpful for the authors. 

1. Introduction lines 62-71: Rather than the biology of gestational malaria, it may be more interesting to use the introduction to delve deeper into the epidemiology of malaria in pregnancy and malaria control measures. For example: have malaria control measures in Uganda changed over time? What are the weaknesses that impair the wide implementation and adherence of the intermittent preventive treatment of malaria in pregnancy in Uganda? How does malaria diagnosis and treatment work in Uganda? 

Author response: Thank you for this comment. We have revised the introduction and the revised sentence reads as follows:

“Malaria is a significant public health problem in sub-Saharan Africa. Of the estimated 228 million global cases of malaria in 2018, 93% were from sub-Saharan Africa. Notwithstanding 94% of the overall 405,000 malaria deaths in the same year were from sub-Saharan Africa. Uganda has one of the highest malaria transmission rates in the region and it is the 3rd largest contributor to malaria global burden according to 2018 estimates. Annually, malaria accounts for 30-50% of outpatient visits and up to one in every five of all reported deaths in Uganda. Of particular concern is malaria in pregnancy (MiP), which is associated with adverse birth outcomes . The less severe but chronic effects of malaria include maternal and newborn anemia, low birth weight, stunting, and reduced cognitive ability in children born to such mothers . The pathophysiological processes preceding adverse birth outcomes in pregnancy are initiated by parasite accumulation in placental intervillous spaces, causing inflammatory reactions and obstruction of nutrient supply to the fetus. Plasmodium falciparum infection, in particular, is associated with very high placental parasite densities and deposition of fibrinoid material. It is this high placental parasitemia and accumulation of immune products that cause placental tissue damage and have the potential to cause adverse birth outcomes including intrauterine fetal growth restriction (IUGR), anemia, spontaneous abortions, preterm labor, low birth weight, and death . 

It is estimated that 10,000 maternal and 100,000 newborn deaths that occur each year in the world are due to malaria. Sub-Saharan Africa alone contributes to 11% of newborn deaths and 20% of stillbirths. Malaria contributes to significant morbidity and mortality in Uganda. It is more common among pregnant women and recent estimates place the prevalence at 15% and the 2019 data indicated that malaria accounted for at least 3500, deaths. The prevalence of MiP from population-based studies in Uganda ranges from 8.9% in the country’s low transmission areas to over 50% in high transmission settings .

The risk factors for MiP are diverse and require context-specific interventions. Current evidence is that the risk of malaria is higher in the first pregnancy, in younger pregnant women and infections tend to peak early and decline towards term reflecting the gradual acquisition of malaria immunity . In a Malawian study, teenage pregnant women (age<20) were more likely to suffer from malaria than older women and the risk was highest in the early trimester. Other risk factors associated with MiP include infection with human immunodeficiency virus (HIV), poor housing, low level of maternal education, and non-adherence to malaria preventive measures like insecticide-treated bed nets and IPTp . 

Uganda has made progress in implementing key malaria control interventions, which include the provision of insecticide-treated bed nets, intermittent preventive treatment for pregnant women, prompt diagnosis and case management (test and treat), and indoor residual spraying in areas with high transmission intensity. The national coverage of insecticide-treated bed nets (ITNs) is reported at 83% while that of IPTp has stagnated at 41%. However, despite these efforts, there is no compelling evidence that the burden of malaria has decreased in recent years. Instead, pockets of malaria ‘epidemics’ have been reported in some parts of the country recently. The major factors that hinder effective implementation of MiP control measures include low utilization of antenatal care (ANC) services (leading to sub-optimal uptake of IPTp), inappropriate case management (leading to malaria rebound or even drug resistance), high transmission intensity, inadequate epidemic preparedness, limited epidemiological and surveillance data to inform decision making.

Unlike in many countries, epidemiological data on malaria in pregnancy in Uganda are scarce despite its public health importance. In 2020, Uganda updated its guidelines for the management of malaria in pregnancy following upsurges in 2018 and 2019. The new guidelines necessitate that all pregnant women attending routine ANC receive a test for malaria. Arua Regional Referral hospital was among the first regional hospitals to begin the implementation of the new guidelines for malaria in pregnancy. We used this opportunity to investigate the magnitude of malaria in pregnancy and its associated factors in this setting”.

2. Methods and materials lines 116 - 117: I suggest specifying whether you included both pregnant women who went to the clinic for routine ANC and pregnant women who went to the clinic because they felt sick. Does the clinic attend women attending for routine ANC and pregnant women with suspected malaria or other diseases? I think it would be interesting to specify the number of ANC visits during pregnancy recommended in Uganda (four?) and the adherence to this recommendation (if you have that information). 

Author response: We did not encounter any symptomatic malaria cases. We enrolled all the women attending the ANC regardless of their morbidity status. The revised sentence reads: 

“We studied 238 women with a mean age of 25.32±5.79 years and all were attending the clinic for routine ANC, no malaria symptomatic cases were recorded”

We have clarified the recommended number of ANC visits in Uganda and the new sentence reads:

“According to the Annual Health Sector Performance Report 2020/2021, only 31% of women attended the recommended four ANC visits and IPTp coverage was below 43.2% in Arua”.

3. Lines 132-133: It is important to specify the Plasmodium species. The diagnosis was done with blood smear microscopy too. 

Author response: We did not use microscopy in this study. We have specified the plasmodium species and the revised sentence reads: 

“Overall, the prevalence of malaria in pregnancy was 26.1 %( n=238) by rapid diagnostic tests (RDT) and 100 %( n=62) of positive cases were due to P. falciparum”.

4. Results line 209: If you have information on the proportion of symptomatic and asymptomatic malaria cases, it may be interesting to include it. 

Author response: Thank you for the comment. We do not have data about symptomatic versus asymptomatic participants as attendance of ANC is not dependent on morbidity. 

5. Lines 146 and 270: Sleeping without insecticide-treated nets on daily basis means no nets, or no nets, or untreated nets? If you have that information it may be interesting to describe it in the discussion. 

Author response: Thank you. We did not collect data about the reasons for not sleeping under insecticide-treated nets and we have acknowledged this as a limitation in the study. Accordingly, we have added a new sentence in the limitations section which reads:

“Second, we did not collect auxiliary data such as reasons for not sleeping under an insecticide-treated net on daily basis among the participants as these were beyond the scope of this study”. 

6. Line 231: I suggest adding one more column with the n of each category (for example "n with malaria/n total"), even though this information has already been placed in table 1. 

Author response: We have added this and details in the new table 

7. Line 306: Regarding the limitations of the study, is it possible that the population most exposed to malaria has less access to the health facility? In this case, this study would not be overestimating the malaria cases. 

Author response: Thank you for this observation. We have removed this limitation and the revised sentence reads;

“There are, however, some limitations to this study. First, our sample size is relatively small and this might have reduced the power to detect a true association in the population. Second, we did not collect auxiliary data such as reasons for not sleeping under an insecticide-treated net on daily basis among the participants as these were beyond the scope of this study. Lastly, we relied on rapid diagnostic tests (RDTs) for malaria screening which might have led to an inaccurate estimate of malaria prevalence”

8. Finally, consider adding a STROBE guideline chart as a supplementary file.

Author response: Thank you for this comment. We have added a STROBE guideline chart as a supplementary file. 

Reviewer #3: 

Summary of the research and overall impression

1. This manuscript aimed to identify the prevalence of MiP and factors associated with MiP among pregnant women in Western Uganda. The introduction should be strengthened with more literature review and presenting the burden of malaria and MiP in Uganda. Methodology and analysis are reasonable. Results session needs revision in text description and background variables analysis should be revisited. The discussion is very weak and required major revision. It needs to improve in terms of approach to discussion, referencing and citation, literature review and linked to the key findings, etc. Academic and scientific English language in presenting and discussing throughout the manuscript is also weak and required proper langue editing.

Author response: We thank you for the wonderful comment. We have addressed the comments as shown in the responses to the below comments.

2. The burden of MiP should be emphasized in the introduction. The overall prevalence of MiP and the burden of malaria in Uganda should be mentioned. 

Author response: We have added a new sentence about the burden of MiP in the introduction section and it reads:

“It is estimated that 10,000 maternal and 100,000 newborn deaths that occur each year in the world are due to malaria. Sub-Saharan Africa alone contributes to 11% of newborn deaths and 20% of stillbirths. Malaria contributes to significant morbidity and mortality in Uganda. It is more common among pregnant women and recent estimates place the prevalence at 15% and the 2019 data indicated that malaria accounted for at least 3500, deaths. The prevalence of MiP from population-based studies in Uganda ranges from 8.9% in the country’s low transmission areas to over 50% in high transmission settings”. 

3. The patho-physiology of MiP describing the inflammatory cellular changes is not mandatory to present in the first paragraph. 

Author response: Thank you for this comment. We have revised the paragraph to highlight events leading to adverse birth outcomes in MiP. The revised paragraph reads: 

“Of particular concern is malaria in pregnancy (MiP), which is associated with adverse birth outcomes. The less severe but chronic effects of malaria include maternal and newborn anemia, low birth weight, stunting, and reduced cognitive ability in children born to such mothers. The pathophysiological processes preceding adverse outcomes in pregnancy are initiated by parasite accumulation in placental intervillous spaces, causing inflammatory reactions and obstruction of nutrient supply to the fetus. Plasmodium falciparum infection, in particular, is associated with very high placental parasite densities and deposition of fibrinoid material. It is this high placental parasitemia and accumulation of immune products that cause placental tissue damage and have the potential to cause adverse birth outcomes including Intra uterine fetal growth restriction (IUGR), anemia, spontaneous abortions, preterm labor, low birth weight, and death”. 

4. Additional literature for the risk factors for MiP should be added. There is only one study referencing for independent risk factors for MiP. 

Author response: Thank you for this observation. We have now added a couple of sentences describing risk factors for MiP and the new paragraph reads: 

“Current evidence is that the risk of malaria is higher in the first pregnancy, in younger pregnant women and infections tend to peak early and decline towards term reflecting the gradual acquisition of malaria immunity. In a Malawian study, teenage pregnant women (age<20) were more likely to suffer from malaria than older women and the risk was highest in the early trimester. Other risk factors associated with MiP include infection with human immunodeficiency virus (HIV), poor housing, low level of maternal education, and non-adherence to malaria preventive measures like insecticide-treated bed nets and IPTp”. 

5. Line 93: please identify the prevalence of pregnant women who present with malaria should be mentioned from the review of routine health facility data, rather than describing “the majority”. 

Author response: Thank you for this comment. We have revised the paragraph and it now reads: 

“Unlike in many countries, epidemiological data on malaria in pregnancy in Uganda are scarce despite its public health importance. In 2020, Uganda updated its guidelines for the management of malaria in pregnancy following upsurges in 2018 and 2019. The new guidelines necessitate that all pregnant women attending routine ANC receive a test for malaria. Arua Regional Referral hospital was among the first regional hospitals to begin the implementation of the new guidelines for malaria in pregnancy. We used this opportunity to investigate the magnitude of malaria in pregnancy and its associated factors in this setting”.

Methods

6. Line 111 (design and setting): pls describe the general of climate setting and specify how the climate setting is described in the manuscript, eg. in figure, or in table, or in appendix. Using “elsewhere” and adding reference (23) is not appropriate. 

Author response: Thank you for this observation. We now have added a paragraph describing the climate setting and how this might influence the malaria pattern throughout the year. The new paragraph reads: 

“Arua sits at an average elevation of 1310 meters above sea level, located between the drier East African plateau and the wetter Congo basin. The climate is generally warmed with an average annual temperature of 23.0 ºC. The warmest month of the year is February with an average temperature of 26.0 ºC and the lowest average temperature in the year occurs in August when it is around 21.2ºC. The rainfall pattern is bimodal with a dry season starting from mid-December to mid-March and a less predictable rainfall season occurs around March to June of every year. The more reliable rainfall season begins in late July until November each year. The warm and relatively humid conditions favor the life cycle of Anopheles gambiae, the main malaria vector in the region. Arua district experiences perennial intense malaria transmission with asymptomatic parasite prevalence rates of 40- 60% in school children. The annual entomological inoculation rate in the region is estimated at 397 infective mosquito bites per person per year with Anopheles gambiae responsible for >80% of infective bites. The peak transmissions occur around August to September and lowest transmission intensities are observed between January and March every year”. 

7. Line 117 (sampling): Why do you use “combination of systematic and simple random sampling”? 

Author response: We first determined a sampling frame, and then employed a lottery method to determine the first and subsequent participants. This process eliminates potential selection bias that might arise due to the non-random selection of first participants and subsequent ones. 

8. Line 120 (sampling): “Women over 10 days” of what? 

Author response: Thank you for this comment. We have clarified this and the new sentence reads:

“Since we enrolled participants over 10 days, we divided the number of women recorded in the ANC register by the number of women required for the interview on each day”.

9. Sampling: It is unclear when the women were enrolled for the data collection, at the time of ANC visit OR during the following visits after sampling approach was applied OR else. Please explain. 

Author response: Thank you for this observation. Our explanation reads:

“We enrolled women attending any of the four ANC visits, at the same visit, blood samples were collected and analyzed”.

10. Data collection: it is assumed, face-to-face interview using paper-based data collection. Please clarify. 

Author response: Thank you for this comment. In the manuscript, we have clarified this and the new sentence reads: 

“We used a paper-based structured questionnaire administered face-to-face by two trained research assistants” 

11. Line 135 (measurement): why do you categorize the age as <25 and >=25 years. 

Author response: Thank you for this observation. We have explained our choice of age cutoff of 25 years and it reads as follows in the manuscript:

“The age cutoff of 25 years was taken as a borderline to compare MiP between early/young women versus mature/older women” 

12. Line 145 (measurement): what is the duration or timepoint defined for “sleeping under bednet” variable? eg. last night or during last week or during last month, etc?

Author response: Thank you for this observation. However, we aimed to assess the consistency of bed net use from the time of conception/pregnancy. 

13. Line 152: the title “statistical issue” is not relevant here. Please delete. 

Author response: Thank you for this observation. The title “statistical issues” has been deleted from the manuscript.

14. Did the measurement variables include any fever or any malaria suggested symptoms at the time of data collection, during ANC visit. If the data is available, pls include the information. Therefore, we can conclude that the cohort of MiP in the study population is symptomatic or asymptomatic. Usually, body temperature should be measured at the hospital during ANC visit; may be for the screening purpose. 

Author response: Thank you for this comment. However, we enrolled women who were attending routine ANC clinics. There were no symptomatic malaria cases, therefore our results represent asymptomatic MiP burden and we have further clarified this in the results section as follows: 

“Since none of the pregnant women had clinical signs/symptoms at the time of presentation, our results represent the burden of asymptomatic MiP”.

15. Ethical consideration: pls mention how the written consent process was approached or applied for those pregnant women who are under-aged (<18 years). 

Results

Author response: Thank you for this observation. We have added a sentence on assent for participants less than 18 years and it reads: 

“For participants below the age of 18 years, an adult either a spouse or guardian provided a written informed consent in addition to assent from the participant”.

16. Line 204: the sentence should not start with the numbers (which is now starting with “173….”). Pls revise. 

Author response: Thank you for this observation. We have corrected this and the new sentence starts like this: 

“One hundred seventy-three, representing 72. 7%(n=238) of the pregnant women…” 

17. Table 1. 

a. It is presenting the “background characteristics of pregnant women and prevalence of MiP”. Pls change the title of the table and in the text. 

Author response: Thank you for this observation. We have corrected this and the new table reads: 

“Table 1: Background characteristics of pregnant women and prevalence of MiP” 

b. N=238 can be mentioned at the top row of the second column, not necessary to include in the first column of each variable. 

Author response: Thank you for this observation. 

We have corrected this and details in the new table 

Discussion 

18. Line 251: using RDT is not “methodologically”, it is about the “diagnostic methods”. Please revise. 

Author response: Thank you for this observation. We have revised the sentence and it now reads: 

“To establish malaria infection, we used rapid diagnostic tests while the previous studies used blood slide microscopy, histology or polymerase chain reaction (PCR)-based molecular methods”.

19. Line 253: what is describe about the limitation of RDT. Pls mention, not just cited to “elsewhere (33)” 

Author response: Thank you for this observation. We have revised this sentence and described the limitations of RDTs as follows: 

“The limitations of rapid diagnostic tests have been described in a recent study and they include low sensitivity compared to microscopy when parasitemia is <100 parasites/µL, a false negative result due to parasite genetic variability, and prozone effect. Additionally, more than 80% of the RDTs used in Uganda are Histidine rich protein (HRP) 2 -based, an antigen-specific for Plasmodium falciparum species. Although this species accounts for ≥95% of malaria infections in Uganda, there is a risk of missing non-falciparum infections”.

20. Line 260: pls spell out “PCR”. 

Author response: Thank you for this comment. We have spelled out the term in full and the revised sentence reads:

“Theoretically, rapid diagnostic tests are inferior compared to blood smear microscopy, placental histology, and polymerase chain reaction (PCR)-based methods”

21. Please discuss why this study used RDT only, not combination with smear microscopy or other diagnostic methods. 

Author response: Thank you for the comment. We relied on RDTs because they are a point of care (POC) test and are used mainly for screening purposes in high-volume clinics in the hospital. In the manuscript, we have added a couple of sentences that read:

“All the participants received a rapid malaria diagnostic test as a standard of care (SOC) blood screening test. Rapid diagnostic tests (RDTs), in this setting, are deployed as a point of care (POC) test partly because of high clinic volumes but also for screening purposes. Therefore, no cases were tested by microscopy or other diagnostic methods”.

22. Line 271-276: the discussion around the use of ITN regarding physical and chemical mechanism is not relevant discussion for the key finding of sleeping under bednet. On the other hand, there is no reference citation for these descriptions. Pls revise the discussion relevant for the use of ITN for this paragraph (line 268-279). 

Author response: Thank you for this observation. We have revised this discussion in the manuscript and it reads: 

“Our data show that malaria in pregnancy is less likely among women who everyday sleep under an insecticide-treated bed net compared to those who never slept under an insecticide-treated bed net every day. This finding is consistent with that of Fana et al in Nigeria and Cisse et al in Burkina Faso. Reports of consistent Insecticide-treated bed net (ITN) use have shown that ITNs are highly effective in reducing malaria incidence. For example, a Malawian study found a 30% reduction in malaria incidence among children less than 5 years after initial household ITN distribution in the study area. In an earlier systematic review, ITNs compared to no nets showed a positive impact on birth weights, and reduced stillbirths/miscarriages by 33%, while placental parasitemia reduced by 23% in all gravidae. It has also been observed that a high coverage and use of insecticide-treated bed nets (ITNs) can potentially reduce mosquito vector density resulting in wider community protection against malaria. However, challenges of misuse of ITNs and lack of proper education regarding care and maintenance for ITNs may reduce their protective effects. For example exposure of ITNs to ultraviolet light (UV), washing them with detergents, and frequency of washing are all important determinants of ITN longevity. Therefore, mass ITN distribution campaigns should be strengthened with the appropriate social behavioral change and communication (SBCC) programs to promote good net use at the household level”.

23. Line 288-289: “Relatedly, the finding that malaria in pregnancy is more likely in the first trimester is consistent with the findings of xxxxxx” is contradict with the key findings from the earlier description of the paragraph. Please check and revise as necessary. 

Author response: Thank you for this observation. However, we meant women who initiated ANC late (i.e in their second or third trimester) were more prone to malaria than those who initiated early enough (first trimester). We have clarified this and revised the paragraph as follows: 

“Our finding that malaria in pregnancy is more likely among women who initiated ANC late (attended first ANC in the second or third trimester) compared to those who attend the first ANC visit in the first trimester is consistent with previous findings of Anchang-Kimbi et al and Nkoka et al . Relatedly, the finding that malaria in pregnancy is less likely in the second or third trimester is consistent with the findings of Simon-Oke et al in Nigeria, Accrombessi et al in Benin, and Gontie et al in West Ethiopia. The biologically plausible reason is the lack of protection against malaria during the first few months of pregnancy due to systematic cytokine bias, which leads to weakened immunity. Early ANC attendance promotes the uptake of malaria prevention services including the use of ITNs and IPTp”. 

24. Age variable should be discussed in the discussion, even though “age” is not statistically significant. It could be due to how the author categorize age (<25 and >=25). Why do you make the cut-off point with 25? Would that be any difference if the author re-categorize the age variable in a different way? It is the demographic characteristics, it should include in the final model, as well.

Author response: Thank you for this important observation. First, we used 25 years as a borderline age to compare MiP between early/young women versus mature/older women. There were few adolescents (age<18) in the study to make meaningful statistics. We have added a paragraph in the discussion that reads:

“Although we did not find a significant relationship between maternal age and MiP prevalence in our study, several studies have reported an association between young maternal age (<20 years) and MiP. The lack of association in our study could be explained, in part, by the fact that a small proportion (18%) of the women studied were below the age of 20 years and this may have reduced the power to detect any significant differences.The association between young maternal age and MiP is thought to be related to the level of acquired malaria immunity which according to literature, increases with number of infections in subsequent pregnancies. This may imply that young women are most likely to be first-time mothers and therefore may have a higher risk of MiP compared to mature women who might have received multiple MiP infections”. 

25. Line 299: what is meant by “modern obstetric care”? 

26. Line 296-304: the discussion in this paragraph is not scientifically sound. Please revise.

Author response: Thank your observations in (25) and (26). However, we found this particular result was confounded by maternal age in our re-analysis of the data. We have removed it from the discussion. However, we have provided an explaination as follows in the results section: 

“The relationship between the variables ‘seeking obstetric care from a TBA and MiP’ was confounded by maternal age in the multivariable analysis”. 

27. Line 307-308: “to the best of our knowledge” and “first few studies in the region” – these descriptions are not proper academic description. And this is not the strength of the study. Pls revise. 

Author response: Thank you for this observation. We have revised the paragraph and it reads: 

“Our study has several strengths and limitations. First, we selected a random sample from a large tertiary care facility which is more likely to be heterogeneous, and hence our results can be generalized to the population under study. Second, we controlled for interviewee response bias by verifying self-reported data against ANC records. Third, we used the malaria indicator survey toolkit which is a validated tool with excellent psychometric properties to collect all the data in the study. There are, however, some limitations to this study. First, our sample size is relatively small and this might have reduced the power to detect a true association in the population. Second, we did not collect auxiliary data such as reasons for not sleeping under an insecticide-treated net on daily basis among the participants as these were beyond the scope of this study. Lastly, we relied on rapid diagnostic tests (RDTs) for malaria screening which might have led to an inaccurate estimate of malaria prevalence. Malaria in pregnancy is often characterized by placental sequestration and therefore reducing parasite load in peripheral blood may result in false negatives. However, there is also a general concern of false negative results associated with the use of rapid diagnostic tests arising from a very high concentration of antibodies or antigens in a specimen, a phenomenon referred to as the prozone effect. Mutational deletions in Plasmodium falciparum Histidine-rich protein 2/3 (pfhrp2/3) gene may also lead to false negative RDT results . However, RDTs are widely used in Uganda as a point of care (POC) test. In limited health care settings, the use of microscopy is limited and often less accurate compared to rapid diagnostic tests. An earlier study in Uganda confirmed this and concluded that RDTs may be more suitable for screening malaria, especially in rural areas. Nevertheless, our results provide useful insights into the problem of MiP in the region and call for the strengthening of malaria control measures among pregnant women”.

Conclusion 

28. Conclusion is weak. Pls revise. 

Author response: Thank you for this observation. We have revised this section and it reads: 

“Our study shows that malaria in pregnancy is high among women seeking ANC at Arua Regional Referral Hospital. The study further shows that consistent ITN use and early ANC initiation may significantly reduce the risk of malaria in pregnancy. We recommend the provision of insecticide-treated bed nets to all pregnant women and early ANC attendance to enable access to malaria preventive therapy and related interventions to prevent malaria in pregnancy”.

In addition, we have summarized key takeaways (implications of the findings) and the new section reads: 

“Our findings have ramifications for clinical care, public health prevention, and control, including surveillance and future research. Our study has shown that MiP is prevalent, necessitating a need for routine surveillance for malaria at the individual and population levels. On the clinical front, our findings emphasize a need to routinely test pregnant women for malaria to enable early diagnosis and treatment of malaria thereby preventing adverse pregnancy and birth outcomes. Our study also provides a benchmark for future research about MiP in the region. To the best of our knowledge, few studies have focused on MiP in the West Nile region so our findings might inform the design of future observational and interventional studies. Policy-wise, the enforcement of malaria prevention and control measures at the family and community levels is important to reduce the burden of malaria in the region. The use of cost-effective and proven malaria prevention measures such as insecticide-treated bed nets (ITNs) and intermitted preventive therapy in pregnancy (IPTp) should be promoted. Environmental control measures such as the draining of stagnant waters on the compound, cleaning/slushing busy areas, and the discarding of broken/empty containers that provide breeding grounds for mosquitoes have to be enforced”. 

We have also included a paragraph of acknowledgement of the support we received from the department to carry out the research and it reads: 

“This study was internally supported by the Faculty of Health Sciences and Department of Nursing and Midwifery and by the investigators”.

---

## [Decision Letter · Decision Letter 1]

2 Dec 2022

PONE-D-22-12136R1High prevalence of malaria in pregnancy among women attending antenatal care at a large referral hospital in northwestern Uganda: a cross-sectional studyPLOS ONE

Dear Dr. Legason,

Thank you for submitting your manuscript to PLOS ONE. After careful consideration, we feel that it has merit but does not fully meet PLOS ONE’s publication criteria as it currently stands. Therefore, we invite you to submit a revised version of the manuscript that addresses the points raised during the review process.

I would like to clarify one point about the role and responsibilities and contributions of the authors in this manuscript and project. I have seen too many first author, senior authors and corespondence author. Need to clariy.

We look forward to receiving your revised manuscript.

Kind regards,

Thae Maung Maung, MBBS, MSc (International Health), PhD

Academic Editor

PLOS ONE

Journal Requirements:

Reviewers' comments:

Reviewer's Responses to Questions

**Comments to the Author**

1. If the authors have adequately addressed your comments raised in a previous round of review and you feel that this manuscript is now acceptable for publication, you may indicate that here to bypass the “Comments to the Author” section, enter your conflict of interest statement in the “Confidential to Editor” section, and submit your "Accept" recommendation.

Reviewer #1: (No Response)

Reviewer #3: All comments have been addressed

2. Is the manuscript technically sound, and do the data support the conclusions?

Reviewer #1: Partly

Reviewer #3: Yes

3. Has the statistical analysis been performed appropriately and rigorously? 

Reviewer #1: No

Reviewer #3: Yes

4. Have the authors made all data underlying the findings in their manuscript fully available?

Reviewer #1: Yes

Reviewer #3: Yes

5. Is the manuscript presented in an intelligible fashion and written in standard English?

Reviewer #1: Yes

Reviewer #3: Yes

6. Review Comments to the Author

Reviewer #1: Malaria in pregnant women is a relevant public health issue. It is a disease that can be prevented, treated and controlled, and therefore, it is unacceptable that there are still maternal or child deaths from malaria anywhere in the world. Therefore, this study is relevant for understanding the burden of disease that malaria in pregnant women still causes.

In the first paragraph of the results, the presentation of absolute numbers and percentages worsened in relation to the initial submission. The suggestion to change the initial submission was to present as an absolute number the research participants with the characteristic that was being presented and, in parentheses, the percentage with a standardized number of decimal places; example: 238 pregnant women were recruited, of which 62 (26.1%) were positive for malaria. The comment that requested correction in the initial submission was motivated by an error that showed the number 238 as corresponding to 76.9% and not to 100.0% of the sample.

In the tables, while the global number is the sum of the different rows in the same column, when stratified between who has and who do not have malaria, the percentages are added in the different columns that make up the same row. This difference in the treatment of data in the same table makes it less understandable.

Regarding the limitations of the present study, this reviewer considers that an important limitation to be discussed is the possibility that the study underestimated the size of the problem if we consider that: 1. The study was recruited during only three months of the year; 2. The study recruited research participants at a time that is not the most prevalent in malaria cases; 3. Prenatal coverage is probably very low at the study site, which allows us to infer that a high percentage of potentially infected pregnant women who do not undergo prenatal care did not have the opportunity to enter the study; 4. The study did not include symptomatic cases; 5. The study made a case diagnosis, with a low sensitivity method, so that some cases could be wrongly classified as uninfected if the parasite density was below the value necessary for the diagnosis. 6. Those who seek prenatal care are likely to be pregnant women who are most protected from malaria and other diseases.

Regarding the determinants, the choice of cutting off the age of analysis in pregnant women under 25 years of age and over 25 years of age concealed the possibility of analyzing adolescence (under 20 years of age) as a risk factor for infection and disease.

Reviewer #3: Thank you for revising the manuscript, addressing all the comments and it reads the scientifically sound paper. Only a few minor comments as followed.

1. The revision for reviewer’s comments was addressed in the authors response, and can be seen in track changes version. But those changes are not accepted in the clean version, meaning the clean version is the same as the original one. It is difficult to read the track change version for the reading of entire manuscript.

2. Please make sure the scientific terminologies are used consistently. For instance, (1) “blood slide microscopy” is not the correct terminology, it should be “blood smear microscopy” throughout the manuscript. (2) malaria parasite species (P.falciparum) should be “italic”.

3. Table 1. Column 3,4. Please revise - MIP “positive” and “negative”, rather than “yes” and “no”.

4. Authorship: in addition to the first author, there are two senior co-authors and corresponding author is the last author. Please clarify “author’s contribution”.

Looking forward to the next revision.

7. PLOS authors have the option to publish the peer review history of their article (what does this mean?). If published, this will include your full peer review and any attached files.

Reviewer #1: **Yes: **Flor Ernestina Martinez-Espinosa

Reviewer #3: **Yes: **Poe Poe Aung

---

## [Author Response · Author response to Decision Letter 1]

16 Dec 2022

Response to Reviewers' Comments 

1. Reviewer #1: Malaria in pregnant women is a relevant public health issue. It is a disease that can be prevented, treated and controlled, and therefore, it is unacceptable that there are still maternal or child deaths from malaria anywhere in the world. Therefore, this study is relevant for understanding the burden of disease that malaria in pregnant women still causes.

In the first paragraph of the results, the presentation of absolute numbers and percentages worsened in relation to the initial submission. The suggestion to change the initial submission was to present as an absolute number the research participants with the characteristic that was being presented and, in parentheses, the percentage with a standardized number of decimal places; example: 238 pregnant women were recruited, of which 62 (26.1%) were positive for malaria. The comment that requested correction in the initial submission was motivated by an error that showed the number 238 as corresponding to 76.9% and not to 100.0% of the sample.

Author response: This error is now corrected and it reads as follows: 

“We studied 238 women with a mean age of 25.32±5.79 years who had attended the ANC clinic; all were not malaria symptomatic cases. Of these, 207 (87.0%) resided within a 5 km radius of the health facility, 191 (80.3%) were married, 149 (62.6%) attained at least a secondary level of education, and 183 (76.9%) were unemployed. Also, 173 (72.7%) were in their second or third trimester, and 117 (49.2%) were pregnant for the first or second time. The majority of the women, 169 (71.0%), started ANC in the first trimester, 71 (30.1%) began IPTp late (> 14 weeks of pregnancy), 137 (57.6%) had adequate knowledge about the correct IPTp doses, 212 (89.1%) reported sleeping under ITNs every day, and 111 (46.6%) reported seeking obstetric care from a traditional birth attendant (TBA) during the current pregnancy. Overall, 62 (26.1%) of the participants had malaria in pregnancy by rapid diagnostic tests (RDT), with all the malaria-positive cases (62 or 100.0%) due to P. falciparum infection. 

In Table 1, our data show that malaria in pregnancy was prevalent in the first trimester compared to the second or third trimester (35.4% versus 22.5%) and among women who attended less than four ANC visits compared to those who had attended ≥4 ANC visits (74.4% versus 71.0%, respectively). However, malaria in pregnancy was less common among pregnant women aged <25 years compared to those aged ≥25 years (25.8% versus 26.3%, respectively), those in the category of one or two pregnancies versus three or more (24.8% versus 27.3%, respectively), those that had ANC visit in the first trimester (≤12 weeks) versus later trimester (>12 weeks) (23.7% versus 31.9%, respectively) and those that had early IPTp initiation (≤14 weeks) versus late IPTp initiation (>14 weeks) (23.6% versus 32.4%, respectively). Being in the first trimester (p=0.044), daily ITN use (p<0.001), and seeking obstetric care from a traditional birth attendant (p=0.036) showed significant differences in malaria in pregnancy in the bivariate analysis.”

2. In the tables, while the global number is the sum of the different rows in the same column, when stratified between who has and who do not have malaria, the percentages are added in the different columns that make up the same row. This difference in the treatment of data in the same table makes it less understandable.

Author response: Thank you for this comment. We have computed and reported row percentages instead of column percentages in the revised Table 1. 

3. Regarding the limitations of the present study, this reviewer considers that an important limitation to be discussed is the possibility that the study underestimated the size of the problem if we consider that: 1. The study was recruited during only three months of the year; 2. The study recruited research participants at a time that is not the most prevalent in malaria cases; 3. Prenatal coverage is probably very low at the study site, which allows us to infer that a high percentage of potentially infected pregnant women who do not undergo prenatal care did not have the opportunity to enter the study; 4. The study did not include symptomatic cases; 5. The study made a case diagnosis, with a low sensitivity method, so that some cases could be wrongly classified as uninfected if the parasite density was below the value necessary for the diagnosis. 6. Those who seek prenatal care are likely to be pregnant women who are most protected from malaria and other diseases.

Author response: We have revised the study strengths and limitations sections along the suggestions and it reads:

“Our study has several strengths and limitations. First, we selected a random sample from a large tertiary care facility which is more likely to be heterogeneous. Hence, our results can be generalized to the population under study. Second, we controlled for interviewee response bias by verifying self-reported data against ANC records. Third, we used the malaria indicator survey toolkit, a validated tool with excellent psychometric properties, to collect all the data in the study. Our study had limitations that should be considered in the interpretation of the findings, and they include 1) A relatively small sample, which might have reduced the statistical power to detect a true association in the population; 2) The study recruitment period was relatively short (October to December or three months), and it coincided with moderate malaria transmission intensity. The malaria transmission intensity in our setting peaks around August and September every year, so the study might have underestimated the burden of malaria in pregnancy; 3) The ANC coverage in our setting was approximately 30.0%, which is lower than the recommended ≥4 ANC visits, implying that a high percentage of potentially malaria-infected pregnant women had missed the opportunity to participate in the study. 

This might have led to an inaccurate estimate of the prevalence of malaria in pregnancy in the study; 4) The study did not include symptomatic malaria cases, and more importantly, there is the possibility that the pregnant women who sought ANC are likely those most protected from malaria and other diseases; 5) The study lacked explanatory variables, such as the reasons for not sleeping under an insecticide-treated net daily among the participants as these were not within the scope of the study; and, 6) malaria in pregnancy was diagnosed with a low-sensitivity method so some cases might have been erroneously classified as uninfected if the parasite densities were below the detection limits. Nonetheless, our results provide valuable insights and should be cautiously interpreted to guide malaria interventions among pregnant women.”

4. Regarding the determinants, the choice of cutting off the age of analysis in pregnant women under 25 years of age and over 25 years of age concealed the possibility of analyzing adolescence (under 20 years of age) as a risk factor for infection and disease.

Author response: We thank the reviewer for this comment. The choice of the age cut-off was informed by the distribution of data, especially the proportion of teenage pregnancies in the study. Our data show that less than 18% of the participants were below the age of 20 years. By using this cut-off, we avoided data sparsity (fewer number of people in each row or column), allowing a more meaningful statistical analysis. We request to maintain the age categories.

Reviewer #3: Thank you for revising the manuscript, addressing all the comments and it reads the scientifically sound paper. Only a few minor comments as followed.

1. The revision for reviewer’s comments was addressed in the authors response, and can be seen in track changes version. But those changes are not accepted in the clean version, meaning the clean version is the same as the original one. It is difficult to read the track change version for the reading of entire manuscript.

Author response: Thank you. We have ensured that the clean and tracked versions of the manuscript are now the same in the revised submission.

2. Please make sure the scientific terminologies are used consistently. For instance, (1) “blood slide microscopy” is not the correct terminology, it should be “blood smear microscopy” throughout the manuscript. (2) malaria parasite species (P.falciparum) should be “italic”.

Author response: We have used “blood smear microscopy” throughout the manuscript, and have deleted “blood slide microscopy”. We have italicized “P.falciparum” in the revised manuscript.

3. Table 1. Column 3,4. Please revise - MIP “positive” and “negative”, rather than “yes” and “no”.

Author response: Malaria in pregnancy has been revised as positive or negative in the tables. Please note that we have not used “MiP” anymore to easy the reading. We have used “Malaria in pregnancy” instead.

4. Authorship: in addition to the first author, there are two senior co-authors and corresponding author is the last author. Please clarify “author’s contribution”. 

Author response: We thank the reviewer for identifying these inconsistencies. We have described the author contributions per the journal requirements. The inconsistencies have been addressed. We now have one corresponding and senior author in the revised submission.

---

## [Decision Letter · Decision Letter 2]

6 Mar 2023

PONE-D-22-12136R2High prevalence of malaria in pregnancy among women attending antenatal care at a large referral hospital in northwestern Uganda: a cross-sectional studyPLOS ONE

Dear Dr. Legason 

Thank you for submitting your manuscript to PLOS ONE. After careful consideration, we feel that it has merit but does not fully meet PLOS ONE’s publication criteria as it currently stands. Therefore, we invite you to submit a revised version of the manuscript that addresses the points raised during the review process.

ACADEMIC EDITOR:

            Expand ITN in the abstract

           Describe the performance of RDT (sensitivity, specificity) used and how the quality of RDT was controlled.

          Sampling technique is not clear. Explain how a combination of systematic and random sampling technique           were used

We look forward to receiving your revised manuscript.

Kind regards,

Musa Mohammed Ali, PhD

Academic Editor

PLOS ONE

Journal Requirements:

Reviewers' comments:

Reviewer's Responses to Questions

**Comments to the Author**

1. If the authors have adequately addressed your comments raised in a previous round of review and you feel that this manuscript is now acceptable for publication, you may indicate that here to bypass the “Comments to the Author” section, enter your conflict of interest statement in the “Confidential to Editor” section, and submit your "Accept" recommendation.

Reviewer #1: All comments have been addressed

Reviewer #3: All comments have been addressed

2. Is the manuscript technically sound, and do the data support the conclusions?

Reviewer #1: Yes

Reviewer #3: Yes

3. Has the statistical analysis been performed appropriately and rigorously? 

Reviewer #1: Yes

Reviewer #3: Yes

4. Have the authors made all data underlying the findings in their manuscript fully available?

Reviewer #1: Yes

Reviewer #3: No

5. Is the manuscript presented in an intelligible fashion and written in standard English?

Reviewer #1: Yes

Reviewer #3: Yes

6. Review Comments to the Author

Reviewer #1: The authors did a great job of proofreading which greatly improved the manuscript. However, the choice that the authors made about the cutoff points of the different exposure variables may limit the comparability of the manuscript with the results of other studies in malaria-endemic locations. For example: having chosen to stratify the population between those younger than 25 and 25 or more years old, as well as having chosen to stratify between first and second pregnancies to compare with more than two pregnancies, limits comparing this with studies that show adolescence and first pregnancy as possible risks for malaria infection. Additionally, it would have been interesting to compare the positivity in the first prenatal evaluation with the positivity of the test in subsequent evaluations, since readers of the manuscript may wonder whether the high prevalence found in the first trimester of pregnancy suggests that, since malaria is such a disease prevalent in the study region, women can start the pregnancy already with parasitemia and who, being diagnosed in the first prenatal evaluation, receive treatment followed by prevention with intermittent preventive treatment, what would make the positivity in subsequent visits decrease?

Reviewer #3: No further comments. All comments were addressed in the revision 2.

Thank you for addressing all the comments for the version 1.

The manuscript is now ready to submit and I accept the revision 2.

7. PLOS authors have the option to publish the peer review history of their article (what does this mean?). If published, this will include your full peer review and any attached files.

Reviewer #1: **Yes: **Flor Ernestina Martinez-Espinosa

Reviewer #3: **Yes: **Poe Poe Aung

---

## [Author Response · Author response to Decision Letter 2]

7 Mar 2023

Comments to the Author

1. Academic Editor comment 1: Expand ITN in the abstract

Author Response: We thank the editor for noticing this error which has now been corrected in the abstract as follows 

“Of the participants, 173 (72.7%) were in their second or third trimester, 117 (49.2%) were first or second-time pregnant women, and 212 (89.1%) reported sleeping under insecticide-treated bednets (ITNs) every day”

2. Academic Editor comment 2: Describe the performance of RDT (sensitivity, specificity) used and how the quality of RDT was controlled.

Author Response: We thank the editor for this insightful comment. About sensitivity and specificity of the test used, we admit this was beyond the scope of this study which has now been added to the study limitations

“Rapid diagnostic assays for malaria based on the histidine-rich protein (HRP-2) have sensitivity and specificity that vary from 95% to 98.2% and 59.2 to 86.3%, respectively, in investigations including general populations. However, due mostly to placental parasite sequestration, the tests are typically less sensitive when used for malaria diagnosis in pregnant women. We admit that despite the fact that it was outside the scope of our study to evaluate the performance of histidine-rich protein (HRP2)-based rapid diagnostic tests (RDTs), this may have contributed to the study's low rate of malaria in pregnant women”. 

Regarding RDT quality control, we have now addressed this under quality control measures as follows 

“The regional laboratory participates in the ongoing national malaria external quality assessment (EQA) scheme and every quarter, a batch testing of RDTs is conducted prior to their issue for laboratory use. The regional external quality assurance team samples RDT cassettes from each batch and are tested using a pair of known positive and negative blood samples confirmed by expert microscopy and PCR. Therefore, the batch used for malaria testing in the present study passed QC testing. The lab technician performed initial testing of the patients and results were reviewed by the research assistant and entered into appropriate data collection forms”. 

3. Academic Editor comment 3: Sampling technique is not clear. Explain how a combination of systematic and random sampling technique were used

Author Response: We thank the editor for this clarification. The revised sentence reads as follows

“We used a combination of systematic and simple random sampling to select the participants. Upon arrival, pregnant women were registered by the midwife, and basic biodata were collected. This information was used to construct a sampling interval. Using previous ANC pattern, we estimated that we would need ten days to enrol 238 participants at an average of twenty-four per day. We divided the number of women recorded in the ANC register by the number of women required for the interview on each day to obtain a sampling interval of five and henceforth, selected every 5th person recorded in the ANC register. Using the registration serial numbers, we then employed a simple random sampling method using a lottery approach to select the first and subsequent participants. This process was repeated daily until the required sample size was reached “. 

4. Reviewer #1: The authors did a great job of proofreading which greatly improved the manuscript. However, the choice that the authors made about the cutoff points of the different exposure variables may limit the comparability of the manuscript with the results of other studies in malaria-endemic locations. For example: having chosen to stratify the population between those younger than 25 and 25 or more years old, as well as having chosen to stratify between first and second pregnancies to compare with more than two pregnancies, limits comparing this with studies that show adolescence and first pregnancy as possible risks for malaria infection. Additionally, it would have been interesting to compare the positivity in the first prenatal evaluation with the positivity of the test in subsequent evaluations, since readers of the manuscript may wonder whether the high prevalence found in the first trimester of pregnancy suggests that, since malaria is such a disease prevalent in the study region, women can start the pregnancy already with parasitemia and who, being diagnosed in the first prenatal evaluation, receive treatment followed by prevention with intermittent preventive treatment, what would make the positivity in subsequent visits decrease?

Author Response: We thank the reviewer for the insightful criticism that has now helped to improve the manuscript. We have addressed the comments on the choice of our cut off for age and gravidity under study limitations as follows

“Not to mention, we recognize the limitations brought on by the choice of our cut-off points for variables like age and gravidity. This limits the comparability with similar studies assessing malaria infection risks from teenage pregnancy and primigravida. Nevertheless, our findings are insightful and should be carefully interpreted to direct malaria interventions among pregnant mothers”. 

The reviewer’s comment on whether women could become pregnant when they already have malaria was very insightful and we have now used it to expand our explanation for the high malaria burden in first trimester as below

“Relatedly, the finding that malaria in pregnancy is less likely in the second or third trimester is consistent with the findings of Simon-Oke et al. in Nigeria, Accrombessi et al. in Benin, and Gontie et al. in Ethiopia. The biologically plausible reason is the lack of protection against malaria during the first few months of pregnancy due to systematic cytokine bias, which leads to weakened immunity. Yet, it is also plausible that women may contract malaria before becoming pregnant, which would explain the increased prevalence seen in the first trimester. It would be interesting to compare malaria positivity at the first prenatal appointment with the positivity of the test during subsequent visits, even if our study did not analyse data from two time points to establish this connection”. 

We also took this opportunity to tidy up the manuscript e.g., the acknowledgement statement now reads like this 

“We thank the pregnant women who participated in the study. We are grateful to the hospital staff, particularly the nurses and midwives working in the ANC clinics who helped with data collection and availed the ANC registers, and made it possible to interview their clients. We also want to express our gratitude to the hospital management for permission to conduct the study”.

---

## [Editor Report · Decision Letter 3]

12 Mar 2023

PONE-D-22-12136R3High prevalence of malaria in pregnancy among women attending antenatal care at a large referral hospital in northwestern Uganda: a cross-sectional studyPLOS ONE

Dear Dr. LegasonThank you for submitting your manuscript to PLOS ONE. After careful consideration, we feel that it has merit but does not fully meet PLOS ONE’s publication criteria as it currently stands. Therefore, we invite you to submit a revised version of the manuscript that addresses the points raised during the review process.

Academic editor: Some minor comments 

Abstract: In this study, RDT was used to diagnose malaria not blood smear microscopy, therefore remove “blood smear microscopy” from the abstract (line #42).

Materials and Methods: in subsection ‘data collection and measurement’ line #179  next to “…. by technician with >10 year experience…”add how RDT was performed in brief  and the performance of the RDT used (sensitivity, specificity,  positivity and negative predictive value)

We look forward to receiving your revised manuscript.

Kind regards,

Musa Mohammed Ali, PhD

Academic Editor

PLOS ONE
---

## [Author Response · Author response to Decision Letter 3]

14 Mar 2023

Comments to the Author

Academic editor: Abstract: In this study, RDT was used to diagnose malaria not blood smear microscopy, therefore remove “blood smear microscopy” from the abstract (line #42).

Author response: We thank the editor for the critical eye into the paper. We have now removed “blood smear microscopy” from the abstract and the new sentence reads like this

“The prevalence of malaria in pregnancy was 26.1% (62/238) by rapid diagnostic testing (RDT)”

Academic editor: Materials and Methods: in subsection ‘data collection and measurement’ line #179 next to “…. by technician with >10 year experience…”add how RDT was performed in brief and the performance of the RDT used (sensitivity, specificity, positivity and negative predictive value)

Author response: We again thank the editor for identifying this omission. We have addressed it and the new paragraph reads like this 

“Briefly, about 5µL of capillary or venous EDTA blood was introduced into the sample well using the sample transfer device that came with the RDT kit. This was followed by two drops of the assay buffer to the assay buffer well. The timer was set for 20 minutes before a test result could be read. The test results were interpreted anytime from 20 but before 30 minutes to minimize inaccurate test results that occur due to reaction product deterioration. A positive result was considered if two lines appeared on the cassette, otherwise it was negative if one control line appeared. In case a test line appeared without a control line or nothing at all appeared, the test was regarded invalid and a repeat test performed on a new device. Standard laboratory practices including aseptic procedures for obtaining samples and infection control measures were observed at all times by the laboratory personnel testing the samples. The performance of the test kits, SD Bioline Malaria Ag Pf (SD Bioline Inc. Korea) were as follows; sensitivity 98.2%, and specificity 91.6%, with positive and negative predictive values of 81.8% and 99.2% respectively”

---

## [Editor Report · Decision Letter 4]

16 Mar 2023

High prevalence of malaria in pregnancy among women attending antenatal care at a large referral hospital in northwestern Uganda: a cross-sectional study

PONE-D-22-12136R4

Dear Dr. Legason,

We’re pleased to inform you that your manuscript has been judged scientifically suitable for publication and will be formally accepted for publication once it meets all outstanding technical requirements.

Kind regards,

Musa Mohammed Ali, PhD

Academic Editor

PLOS ONE
---

## [Editor Report · Acceptance letter]

27 Mar 2023

PONE-D-22-12136R4 

High prevalence of malaria in pregnancy among women attending antenatal care at a large referral hospital in northwestern Uganda: a cross-sectional study 

Dear Dr. Legason:

I'm pleased to inform you that your manuscript has been deemed suitable for publication in PLOS ONE. Congratulations! Your manuscript is now with our production department. 

Kind regards, 

on behalf of

Dr. Musa Mohammed Ali 

Academic Editor

PLOS ONE